# Architectural Continuity Assessment of Rural Settlement Houses: A Systematic Literature Review

Xiaokang Wang [1], Li Zhu [2,*], Jiang Li [2], Ni Zhang [2], Yue Tang [2], Yilin Sun [1], Honglin Wu [2] and Chuang Cheng [2]

[1] Research Center of Chinese Village Culture, Central South University, Changsha 410083, China; 220501002@csu.edu.cn (X.W.); 210501002@csu.edu.cn (Y.S.)

[2] School of Architecture and Art, Central South University, Changsha 410083, China; lijiang@csu.edu.cn (J.L.); 221301003@csu.edu.cn (N.Z.); 211301005@csu.edu.cn (Y.T.); 221311060@csu.edu.cn (H.W.); 211311066@csu.edu.cn (C.C.)

[*] Correspondence: 207131@csu.edu.cn

**Abstract:** As a significant component of rural settlements, residential architecture is a record of historical changes containing considerable research value. In the study of residential architectural continuity, the focus is on the inheritance and innovation of traditional residential architectural "genes" in contemporary new residential buildings. Based on a systematic review of the literature, the purpose of this study is to analyze the research trends, categories, and variables relating to architectural continuity in residential buildings, and to build a systematic and comprehensive framework for assessing the architectural continuity of residential buildings in rural settlements based on prior research. This study provides guidance and references for evaluating the design of new residential buildings in rural settlements and for formulating regional planning principles. Using the PRISMA guidelines as the basis for the review method, we filtered the literature from three databases: Web of Science (WoS), Scopus, and EI, and studied the 40 articles selected at the end. As can be seen from the results, the literature on architectural continuity in rural settlements has focused more on functional and typological levels and less on the archetypal level of architectural continuity (user behavior). Still, the archetypal level is the most important, because the behavior of housing users directly influences the degree to which housing is continuous in terms of its function and type. The most important finding of this review is that the resident behavior of housing users has a significant influence on the assessment of the continuity of housing architecture, and thus, the continuity of housing architecture in rural settlements must be assessed in conjunction with user behavior.

**Keywords:** rural settlement; residential architecture; continuity; systematic literature review

## 1. Introduction

Vernacular architecture represents cultural patterns and lifestyles. It is closely related to space, time, and materiality [1]. In the course of its development, the user makes the necessary changes according to social and environmental constraints in order to adapt them to the actual needs of each place [2]. Residential architecture is a major component of vernacular architecture [3]. An analysis of 127 studies over the past 30 years shows that the number of studies on vernacular architecture has increased dramatically since 2007 [4], demonstrating the significant potential of research on the continuity of residential architecture. But continuing the traditional architectural features in the new houses isn't easy [5]. In 1999, the International Council on Monuments and Sites (ICOMOS) adopted the Charter for the Heritage of Vernacular Architecture, which explicitly stated the need to maintain architectural coherence through the continuity of architecture [2]. Therefore, it is essential to investigate the continuity of residential architecture.

The Oxford Dictionary defines the word "continuity" as "the persistence or operation of something over time and space" that has two layers of connotation. On one

hand, "continuity" can be described as the preservation, maintenance, or protection of an object, emphasizing that the original object is preserved in the objective spatial environment as time changes. On the other hand, continuity can be more focused on persistence, specifically on the objective level of objects, especially residential buildings, which are affected by changes in time, space, and human activities, and will eventually disappear one day. The word "continuity" can then be understood as the ongoing existence or functioning of content, the inheritance and continuation of the architectural characteristics of residential buildings, which is similar to the concept of "morphogenetic", which originated in the field of biology. It is also important to note the following: in this paper, we propose traditional residential architecture that refers to architecture with regional features, constructed using local materials and traditional craftsmanship in the context of conventional production and life, reflecting residents' social habits and cultural values [6]. In general, continuity in residential architecture is no longer seen as simply preserving what remains of the past, but rather as concentrating on inheriting and developing the unique features of traditional residential buildings in contemporary new residential buildings [7].

Architectural continuity has been extensively studied. The famous architect Bruno Zevi discussed the study of the spatial continuity of Gothic architecture in his book "Architecture as space" as early as 1993. He argued that the spatial continuity of Gothic architecture is the result of the dynamic development of society, and that the degree of continuity represents the vitality of the architecture [8]. Although residential architecture represents local culture and tradition, it does not necessarily mean it is 'timeless' or 'unchanging' [7]. Influenced by socio-economic patterns and lifestyles, "change" is accepted as part of the continuity of residential architecture [9]. Rural settlements maintain their continuity in the process of constant change, and they pass this on to the next generation through social production, adoption, and transmission [10–12]. The continuity of residential architecture in rural settlements is a dynamic concept closely related to social change and development. The discussion on the continuity of residential architecture is more about the recreation of the dynamic evolution of residential architecture; it is also the process through which there is a continuance of its unique identity.

However, in recent years, the residential architecture of rural settlements has been influenced by the interregional spread of culture and global socioeconomic transformation [10]. Both new residential architectures in the same region and traditional residential architectures face the problem of style discontinuity. Scholars have approached this question from two perspectives. Researchers have focused on assessing the continuity of architectural forms. Several studies point out that new residential houses in villages entirely mimic the forms of urban architectural design, neglecting regional characteristics and causing the problem of formal discontinuity between new and traditional residential homes in rural settlements [13–17]. In terms of this level of research, researchers primarily use quantitative analysis to compare the similarity and consistency of the objective forms of traditional and new housing, and then assess the degree of housing continuity. On the other hand, many researchers focus on assessing the continuity of architectural culture in residential buildings [18–21]. Various regional, ethnic, and religious cultures influence the creation and continuity of the residential architecture. Qualitative analysis is used to analyze the cultural aspects that influence residential architecture, with the researcher making subjective judgments about the degree of continuity of the residential architecture, and offers corresponding suggestions for improvement.

The above two aspects of research allow us to learn that in the process of continuity of residential architecture and because of the different focuses of research, the resulting assessment of continuity of residential architecture is often limited to one level in the absence of a comprehensive evaluation of continuity of residential architecture. There are differences in research methods and variable selection, which hamper the assessment of the continuity of residential architecture to some degree. Thus, this study aims to construct a systematic and comprehensive framework for assessing the architectural continuity of rural settlements based on previous studies, which is of practical importance for the development

of planning principles and maintenance of the architectural features of rural settlements. To this end, this study asks the following questions: (1) What are the trends in studying the architectural continuity of rural settlement housing?, (2) Which category is the most important for studying architectural continuity in rural villages? What is the reason?, and (3) What variables are involved in the study of architectural continuity in rural villages? A systematic review of the literature on the subject of architectural continuity of rural settlements would allow us to quantify the importance of the methods and variables used in the assessment of architectural continuity in residential buildings. It would also enable us to construct a systematic and comprehensive framework for assessing the architectural continuity of rural settlements from both architectural character and resident behavior perspectives, based on the points of view of previous studies, which may help the country or region inherit regional features and maintain diversity in architectural culture.

## 2. Materials and Methods

This study followed a systematic literature review process adapted from that of Boland et al. (2017) [22]. A protocol for searching and screening articles to minimize bias and ensure the accuracy and comprehensiveness of the data was developed [23]. A systematic literature review enables a comparison of inconsistencies in the literature on a particular research area to guide decision-making [24] and to identify future research directions as well as research frameworks [25–27]. Following the preferred reporting items for systematic reviews and meta-analyses guidelines, keywords relevant to this study were identified. Next, the process of literature selection and collection is described; that is, searching the literature through databases and performing initial refinements and exclusions within the databases. Finally, by developing an inclusion and exclusion criteria, the literature derived from the database was rescreened using Rayyan (a web and mobile app for systematic reviews) to determine the final number of studies to be included [28]. Rayyan helps to rapidly screen the literature; it is a semi-automated way of handling documentation and allows multiple individuals to work together through networks, which is an obvious advantage compared to the literature screening tools used in the systematic literature review. The Scopus, Web of Science, and EI databases were selected because they contain a wide range of data [29] and are peer-reviewed with high authority, ensuring the quality of the literature.

### 2.1. Keyword Selection

Firstly, we entered the keywords "residential", "continuity", and "village" into the three databases for the initial search. We found one, six, and five articles related to the keywords, respectively, but they were considered invalid because these numbers of articles returned were rather insignificant. To expand the results, we added search terms related to this study by referring to keywords in the systematic literature review related to residential architecture to ensure that more literature could be obtained [30,31]. Next, to address the issue of selecting keywords for the search, we conducted a discussion with experts in the field of residential architecture research within the group and, in the process, added keywords related to the study of the continuity of residential architecture, such as vernacular architecture, similarity, and inheriting. Finally, we used synonyms for "residential architecture", "continuity", and "rural" and used the wildcard "*" to construct our search formula for a comprehensive literature search (Table 1).

**Table 1.** Keyword selection.

| Residential Architecture | Continuity | Countryside |
| --- | --- | --- |
| Vernacular architecture | Continuity | Village |
| Traditional courtyard | Consistency | Countryside |
| Residential | Similarity | Rural |
| Dwellings | Inheritance | |
| House | Cultural sustainability | |

## 2.2. Literature Search

We searched three databases, beginning in April 2023, to identify the literature required for a residential architecture continuity review. The literature search formula (Table 2) was constructed using keywords from Part One, and a literature search was conducted through 2022 to ensure the reproducibility of the study [32]; the language of the literature was limited to English to ensure that the expression of ideas in the literature could be accurately judged in the next stage, namely reading the literature to avoid misunderstandings. We searched the Web of Science database by selecting the citation indices SCI, SSCI, and AHCI from the core collection, which resulted in 563 articles. Since the search process involved keyword matching only, a large number of papers outside the scope of the present study were obtained, which had to be initially screened using the filter bar on the left-hand side of the database for articles unrelated to this study. A total of 117 articles remained after this screening process. This search form was imported into the Scopus database, which yielded 1151 articles, and after excluding the categories of topics unrelated to this study, 359 articles remained. The same search form was entered into the EI core database, yielding 1050 articles, and after excluding the articles unrelated to our search domain by "controlled vocabulary", 407 articles remained. Our initial screening yielded 883 studies. The acquired literature was exported to the database as the basis for the second literature screening.

**Table 2.** Database search build.

| Database | Search Type | Result |
|---|---|---|
| WoS | (TS = ("Vernacular architecture" OR "residential" OR "Traditional-Courtyard" OR "dwellings*" OR "house*")) AND (TS = ("Cultural sustainability" OR "continuity*" OR "consistency*" OR "Similarity*" OR "inheriting")) AND (TS = ("Village*" OR "countryside* OR "rural")) | 117 |
| Scopus | (TITLE-ABS-KEY ({vernacular architecture} OR {traditional courtyard} OR residential* OR dwellings* OR house*) AND TITLE-ABS-KEY (continuity* OR consistency* OR similarity* OR {cultural sustainability} OR inheriting) AND TITLE-ABS-KEY (village* OR countryside* OR rural*)) | 359 |
| EI | (((((Vernacular architecture or Traditional Courtyard or residential or dwellings or house) WN KY) AND ((Cultural sustainability or continuity or consistency or Similarity or inheriting) WN KY)) AND ((countryside or village or rural) WN KY))) | 407 |

## 2.3. Literature Screening

Given that the 883 articles screened in the database were only obtained by keyword matching and that there were different research topics within them, we had to obtain more literature that corresponded to the purview of the current study by importing it into Rayyan, a web and mobile app for systematic reviews, for a second screening (based on title and abstract) [28]. First, to save time in the screening process, it was necessary to check for duplicates of these 883 articles before the secondary screening. Because different databases duplicate the same articles, the literature with high similarity was listed by the duplicate filter command of the Rayyan application, after which 64 duplicate articles were deleted, leaving 819 articles. Second, we performed a second screening of the literature based on the developed inclusion and exclusion criteria (Table 3), combined with the literature titles and abstracts. Rayyan provided a visualization of titles and abstract content so that we could read it more easily, and we obtained 66 articles after the second literature screening. We then had to conduct a third screening (full-text reading) to ascertain whether the 66 articles we had screened were closely linked to the concept of residential continuity that our current study was exploring. For the full-text reading process, the exclusion and inclusion criteria were based on whether the articles addressed continuity in residential

architecture. In the end, after the third literature filtering, we obtained 29 papers directly related to the continuity of residential architecture in rural settlements. To ensure the completeness of the study, we also included 11 additional articles through the snowball literature screening approach, and 40 articles were ultimately identified as the sample for the present study. The flow diagram of the literature search and screening is depicted in Figure 1. The remaining authors participated in a third full-text screening of the 66 articles to increase the validity of the study. Sixty-six articles were assigned to the "included", "excluded", or "unsure" category. The reasons for uncertainty were recorded in writing, and the final decision on inclusion was reached through a group discussion to address the areas of controversy. Furthermore, to ensure rigor in the included literature and to invite experts (This research group) to sample the quality of the third literature search from the final 40 screened articles, experts within the subject pool selected one piece out of five for full-text reading to determine the accuracy of the list included in the literature.

**Table 3.** Inclusion and exclusion criteria.

| Inclusion Criteria | Exclusion Criteria |
| --- | --- |
| Residence | Non-residence |
| Related to continuity | Nothing to do with continuity |
| Rural area | Non-rural areas |

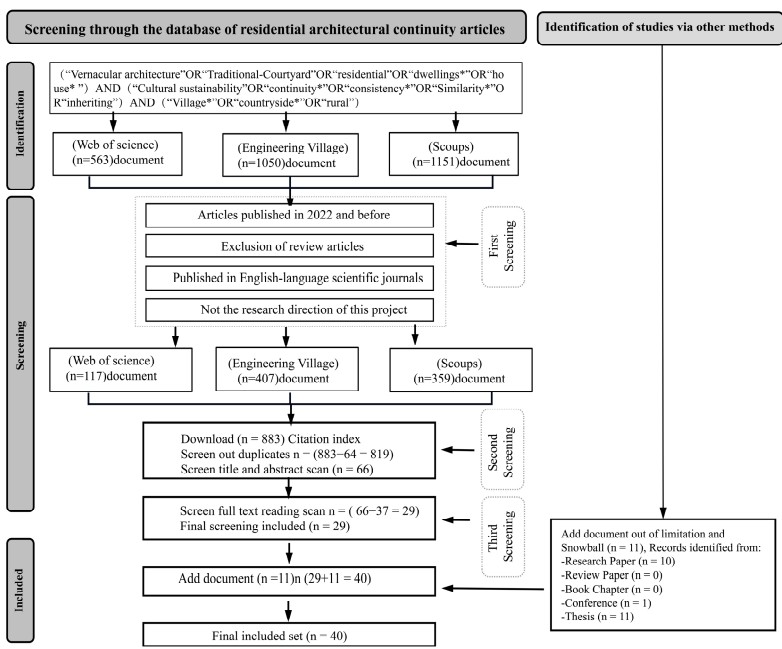

**Figure 1.** PRISMA flow diagram for the article selection process.

## 3. Results

### 3.1. Tendencies in the Study of the Continuity of Residential Architecture

Table 4 shows that most studies of architectural continuity in rural settlements were conducted in developing countries in the eastern hemisphere, with the highest number of studies being conducted in China. This is because villages are gradually being incorporated into urban areas with the expansion of cities in developing countries. In that process, traditional housing faces the problem of conservation and development [33], which also demonstrates that research on the continuity of housing architecture needs to pay greater attention to the developing world.

**Table 4.** Article of the included studies based on PRISMA statements.

| Study ID | Title/ Author | Region | Continuity Classification | Constant | Model/Theory/ Methodology | Nature of Research |
|---|---|---|---|---|---|---|
| 01 | [14] | / | continuity of type | culture | Recog-Net | quantitative and qualitative combination |
| 02 | [34] | Indonesia | continuity of archetypal | custom culture values belief | Mimesis-semiotics method | qualitative analysis |
| 03 | [35] | Algeria | continuity of type | culture values custom lifestyle | Typological analysis Spatial syntax gamma analysis | quantitative analysis |
| 04 | [36] | Iran | continuity of type | culture | Mathematical relations Python | quantitative and qualitative combination |
| 05 | [33] | Poland | continuity of archetypal | culture lifestyle custom | Variance clustering | quantitative and qualitative combination |
| 06 | [37] | China | continuity of type | culture custom memory belief | SPSS Matlab | quantitative analysis |
| 07 | [38] | Palestine | continuity of function | culture | EDSL | quantitative analysis |
| 08 | [39] | China and Poland | continuity of function | culture custom concept | / | qualitative analysis |
| 09 | [40] | Marshall | continuity of archetypal | culture value belief | / | qualitative analysis |
| 10 | [21] | India | continuity of type | social culture | Justified Floor Plan Spatial syntax | quantitative and qualitative combination |
| 11 | [41] | / | continuity of type | culture concept | / | qualitative analysis |
| 12 | [42] | Turkey | continuity of archetypal | culture memory | Plot comparison data comparison | qualitative analysis |
| 13 | [17] | Italy | continuity of type | values culture lifestyle | Clustering principal component analysis | quantitative analysis |
| 14 | [43] | Indonesia | continuity of function | philosophy culture values | life-cycle analysis | quantitative analysis |
| 15 | [44] | Cyprus | continuity of function | philosophy | Mahoney table method | quantitative and qualitative combination |
| 16 | [45] | China | continuity of function | cognitive | / | qualitative analysis |
| 17 | [46] | Turkish | continuity of type | culture social values | Shape syntax | quantitative analysis |
| 18 | [47] | China | continuity of function | culture | SI | qualitative analysis |
| 19 | [48] | Turkey | continuity of archetypal | culture values custom memory | / | qualitative analysis |
| 20 | [49] | / | continuity of type | culture lifestyle | / | qualitative analysis |
| 21 | [50] | Bulgaria | continuity of function | / | / | qualitative analysis |

| Study ID | Title/ Author | Region | Continuity Classification | Constant | Model/Theory/ Methodology | Nature of Research |
|---|---|---|---|---|---|---|
| 22 | [51] | China | continuity of function | culture rules habit values | / | quantitative analysis |
| 23 | [5] | Thailand | continuity of function | / | Analytic Hierarchy Process\Multiple criteria Decision Making \Similarity to Ideal | quantitative and qualitative combination |
| 24 | [52] | China | continuity of function | philosophy custom culture | / | qualitative analysis |
| 25 | [53] | Egypt | continuity of type | culture values | / | qualitative analysis |
| 26 | [54] | China | continuity of type | custom culture | / | qualitative analysis |
| 27 | [55] | China | continuity of type | culture custom | / | qualitative analysis |
| 28 | [56] | China | continuity of function | custom belief values habit behavior | / | qualitative analysis |
| 29 | [57] | Cyprus | continuity of archetypal | environment culture value lifestyle | / | qualitative analysis |
| 30 | [58] | Turkey | continuity of type | culture belief values environment memory | / | qualitative analysis |
| 31 | [12] | Indonesia | continuity of archetypal | Rules values culture belief lifestyle | / | qualitative analysis |
| 32 | [11] | Indonesia | continuity of archetypal | environment | naturalistic paradigm | qualitative analysis |
| 33 | [59] | Indonesia | continuity of type | values lifestyle | / | qualitative analysis |
| 34 | [60] | Algeria | continuity of type | culture | Spatial syntax gamma analysis | quantitative and qualitative combination |
| 35 | [61] | Turkey | continuity of type | environment culture memory emotion | / | qualitative analysis |
| 36 | [1] | / | continuity of function | motion culture values memory | / | qualitative analysis |
| 37 | [62] | China | continuity of type | memory emotion culture | / | qualitative analysis |
| 38 | [21] | Marshall | continuity of archetypal | culture values memory belief lifestyle | Spatial syntax | quantitative and qualitative combination |
| 39 | [63] | Itcaly | continuity of type | culture social | Variance clustering | quantitative and qualitative combination |
| 40 | [64] | Sri Lanka | continuity of function | philosophy values | / | qualitative analysis |

In terms of the trends in research methods on the continuity of residential architecture, a visual analysis was conducted on the techniques, models, and theories used in the continuity of residential architecture in the included literature. As shown in Table 4, the most prominent feature of the methods used in exploring the continuity of residential buildings was cross-use. A few studies have been conducted in one way, and the cross-use of methods can explain and analyze the research objects more clearly. Space syntax, gamma analysis, cluster analysis, and variance are the methods used to study the continuity of residential buildings. Space syntax is the most frequently used because it mainly analyzes the spatial organization of the study, which is also the most commonly used content in research on the continuity of residential buildings. On the other hand, the spatial syntax method (a new language for describing architectural and urban spatial patterns) focuses on analyzing the spatial organization of the building (plan). However, other variables in the assessment of the continuity of residential buildings, such as material, color, and structure, should be compared using cluster analysis and the variance method. Cluster analysis and the variance approach can analyze multiple variables involved in residential building continuity and rank residential building similarities (using formulas) to compensate for the lack of research on spatial syntax [33]. As shown in Table 4, 24 out of the 40 articles were included in the qualitative analysis. In recent years, the number of qualitative and quantitative studies has increased. Thus, the framework for assessing the continuity of residential buildings constructed in this study was a combination of quantitative and qualitative studies. We did not plan to use a single method, but a combination of the spatial phrase method, gamma analysis, cluster analysis, and variance method, as needed.

### 3.2. Research Classification of Continuity of Rural Settlement Dwellings

The "deep beauty" framework proposed by Professor Gary J. Coates for local architecture research was used for reference in the classification of residential architecture continuity research. This framework is primarily composed of three interlinked levels: functional, typological, and archetypal. Research papers on the continuity of residential architecture can be categorized in terms of the different emphases on the three levels of the "deep beauty" framework. It also forms the basis of the framework for assessing the continuity of residential architecture in the present study, namely the continuity of function, type, and archetype (Table 4).

"The functional level includes design for all the pragmatic needs of the building's users. Truly functional buildings are also artfully integrated with their sites and respond simply and appropriately to the available sunlight, wind, and light. Such buildings, which are always no larger than they need to be, are necessarily energy efficient" [65]. Residential buildings continue to function primarily through the extraction of environmentally friendly technologies, such as ventilation, thermal insulation [52], and thermal insulation of traditional residential buildings in rural areas. By combining them with modern technologies and applying them to contemporary new residential buildings to alleviate the problem of transient energy consumption in modern construction technologies while improving the comfort of residential buildings, the continuity of the function of traditional residential buildings is realized. Many studies have proven that learning about ecological construction experience (use of renewable materials, the selection of appropriate wall thicknesses for insulation, and the selection of appropriate window and door sizes) from traditional dwelling building functions is no longer an option [5,13,14], but a must [66]. In contrast, the continuity of residential building functions should not only focus on energy savings, renewable resources, or building technology but should also consider the psychological will of residential users [67]. A unilateral focus on building sustainability tends to ignore the actual needs of users.

"The typological level involves the adaptation of bio-regional building traditions and historically situated building types in the design of contemporary buildings that are capable of evoking a sense of connection with history, community, nature, and place" [65]. By continuing the type of residential building, something, both familiar and unfamiliar,

is created by redesigning the form of the residential building to fit the characteristics of the time [37,49,53,55,62,63]. Residential architecture is an ever-changing entity with transformations in time and space [1]. In addition, the continuation of residential building types is similar to the concept of "morphogenesis". Extracting morphological features from traditional residential buildings, transforming them, and applying them to new contemporary residential buildings, can evoke user connections to the area, enhance residents' sense of identity for new contemporary residential buildings, and play a significant role in continuing residential buildings.

"The archetypal level is the deepest layer of meaning and metaphorical significance in architecture. Buildings that reach this level lead users back through layers of consciousness and time from the outer surface of the waking mind to the depths of what Carl Jung calls the collective unconscious" [65]. The study of the archetypal dimension of the continuity of residential architecture involves the collective unconscious, which Jung argues consists of archetypes and is a model of instinctive behavior [68]. In other words, the archetypal dimension of continuity in residential architecture focuses on the behavior of residential users. Residential user behavior reflects cultural values, akin to the modern Western emphasis on "persons with a sense of continuity and tradition do not need to preserve the past [69]". Therefore, research on the archetypal continuity of residential buildings must focus on the behavior of users in residential spaces.

In summary, the functional level focuses on energy conservation and energy sustainability, which can be summarized as a study of building applicability; the typological level focuses on the redesign of morphological elements in residential building types, which is a continuation of the regional characteristics of residential buildings; and the archetypal level emphasizes that the formation of residential buildings is the presentation of the collective unconscious, which is influenced by cultural values and it impacts the continuation of residential buildings through the behavior of residential users. Following the "deep beauty" framework proposed by Professor Coates, the three levels of function, type, and archetypal focus on different study aspects. The 40 included articles were categorized, revealing the following (Figure 2). It can be seen that the research literature on the continuity of residential architecture is predominantly focused on the level of type, followed by the function level, and there is a relatively low focus on the archetypal level because research on the continuity of residential architecture is mainly conducted from the perspective of architectural design, with a focus on exploring the objective elements of the architecture, and it easily ignores the influence of user behavior on residential space. In recent years, however, there has been a growing trend in research into the continuity of residential architectural archetypal, and the research approach is primarily qualitative, with a focus on residential resident behavior [17].

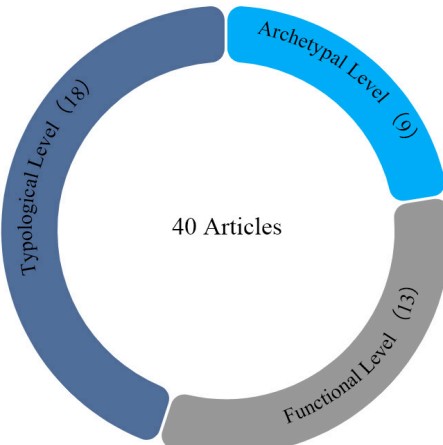

**Figure 2.** Classification of residential building continuity.

### 3.3. Variables of Continuity in Residential Architecture

The residential building continuity variable was divided into two levels: architectural characteristics and resident behavior. Extracting the continuity variables from the literature (Table 5) allowed us to determine their importance in the continuity study process. This laid the groundwork for selecting variables for a residential building continuity assessment framework. The architectural character variables were ranked according to the number of variables involved in the study of the continuity of residential architecture (top 10) in the order of materials [33], plan [35], technology [45], morphology [36], decoration [33], color [17], volume [17], structure [47], façade [48], and size [51]. Materials are the most commonly used construction variable because they are the most rapidly updated and are closely related to advances in construction technology. Resident behavior variables were ranked according to the number of variables involved in the study of the continuity of residential architecture (top 10) in the following order: construction [43], cooking [46], rites [56], parties [39], sleep [40], socializing [8], maintenance [56], storage [49], rest [36], and feeding [46]. Construction, cooking, and ritual behaviors are also the primary focus of behavioral research because they are related to the everyday lives and spirituality of residential consumers. Through the study of residential behavior, we can understand, to a great extent, the reasons for ecological technology at the functional continuity level of residential buildings. The study of culinary behaviors can reveal the roots of the family rituals of residential users, and ritual behaviors are related to the production of certain special elements, spaces, and forms in residential spaces. In addition, the variables of architecture and behavior involved in the continuity of residential architecture intersect in the study of functions, types, and archetypes. Moreover, they are not limited to a single level, which also shows that any variable in the residential space is influenced by many aspects, and that the evaluation of the continuity of residential architecture needs to be viewed systematically and holistically.

**Table 5.** Continuity of residential architecture variables.

| Building Variables | Study ID | Number | Behavior Variables | Study ID | Number |
|---|---|---|---|---|---|
| material | 01 04 09 10 12 15 16 21 23 26 31 32 34 37 | 14 | construction | 09 11 13 14 15 21 22 24 25 26 | 10 |
| plane | 03 08 12 16 24 25 31 32 37 34 | 10 | cooking | 04 09 10 12 16 20 29 31 37 | 9 |
| technology | 04 13 22 25 29 31 32 37 | 8 | rite | 02 09 22 31 32 36 37 | 7 |
| morphology | 02 04 10 16 20 22 | 6 | party | 03 08 09 10 16 25 33 | 7 |
| decoration | 02 09 14 31 36 | 5 | sleep | 04 09 10 12 23 29 37 | 7 |
| color | 09 16 29 32 34 | 5 | socializing | 03 10 25 31 32 34 | 6 |
| volume | 11 20 25 38 | 4 | maintenance | 09 12 13 15 22 | 5 |
| structure | 02 03 12 31 | 4 | Storage | 02 10 20 38 | 4 |
| facade | 08 10 13 33 | 4 | rest | 04 31 32 34 | 4 |
| size | 15 38 | 3 | feeding | 10 12 25 34 | 4 |

## 4. Discussion

Globally, research on architectural continuity can be traced back to the study of urban morphology in England, Italy, and France. The British Conzenian school combined urban planning with research on urban morphology and introduced the concept of morphogenesis into the field, focusing on studying the morphology of the built environment during historical change. In contrast to the British school, the study of Italian urban morphology was closely integrated into the practice of architectural design in cities from an early stage, with a focus on combining local features of buildings in the design process and an emphasis on the continuity of the inherent laws of architecture (the law of space organization) [70]. This also played a role in guiding the conservation of historic architectural heritage in this period. The French studies of urban morphology, influenced by the Italian school, share similarities with it, and it focuses on the dialectical relationship between the physical

development of cities, types of residential buildings and residential building design from the point of view of topological–geometric relationships, with a focus on "pure" morphology [71]. Therefore, the history of early urban morphology in Western countries shows that, although there is sufficient research on the analysis of architectural forms from the perspective of urban morphology, research on architectural continuity is mainly based on objective physical aspects. With this in mind, the research in this study will supplement the user behavior factors that affect the continuity of the architecture of rural settlements.

The categorization of the 40 papers in the study revealed that the number of studies addressing the archetypal level of continuity of residential architecture is lower than the functional and typological levels, but they are the most significant ones because the extraction of constants (extracted through articles) in the study of residential building continuity reveals (Table 4) that the functional, typological, or archetypal levels of residential building continuity involve factors, such as culture, values, customs, and memories, which influence the creation of residential buildings through the behaviors of users. It is thus clear from the analysis above that the behavior of residential building users, which is emphasized at the archetypal level, is an essential part of assessing the continuity of residential buildings, and the most important finding of this review is the construction of a combined architectural character and resident behavior framework for assessing the continuity of residential buildings by considering the behavior of users as a component of the continuity assessment framework of residential buildings. Here, we use both architectural character and resident behavior terms because we wish to show that the continuity of residential buildings is influenced not only by a single level of objective (architectural form aspect) but also by residential user behavior, which affects the continuity of residential buildings. Thus, the evaluation framework constructed in this study includes both levels.

Traditional residential architecture, as a continuous creation of life, is composed of three interlinked levels: function, type, and archetypal. First, from the functional level of residential buildings, although passive ecological technologies can meet the needs of contemporary space users to a certain extent compared with active cooling and heating technologies, active technologies break the limitations of the climatic environment for residential buildings, which provide obvious comfort and dynamism, and it is problematic to explore continuity only at the functional level. Second, at the type level, residential material variables account for the largest proportion of the research literature on the continuity of residential buildings. However, the continuity of residential buildings faces difficulties in providing traditional building materials (both subjective factors, such as government restrictions, and passive factors, such as the disappearance of resources or environmental damage) and the loss of knowledge of traditional construction techniques, which result in certain obstacles in the continuity of residential buildings. The emergence of new materials and technologies that are faster and cheaper, and more convenient construction methods, have consistently impacted the continuity of original building types. Finally, the continuity of the architectural archetype of residential buildings emphasizes the behavior of the residential users. In researching housing function and type, scholars have noted that the degree of continuity in the function and type of traditional housing is influenced not only by the local environment, but is also linked to resident behavior [53,72]. In other words, the evaluation of the continuity of rural settlement housing architecture must combine three levels: continuity of function, continuity of type, and continuity of the prototype.

First, based on the variables obtained from residential building continuity in Table 5, the ten most critical architectural variables were selected in the order of material, plan, technology, morphology, decoration, color, volume, structure, façade, and size, according to the coding arrangement from A-1 to A-10. In the process of field research on the continuity of residential buildings, we can use photography, mapping, and aerial photography to collect the objective aspects of the continuity of residential buildings as the primary data of the continuity of residential buildings. Furthermore, at the behavioral level of continuity of residential buildings, top ten variables were selected based on their importance in the following order: construction, cooking, rites, parties, sleep, socializing, maintenance,

storage, rest, and feeding. These variables were used as critical components of the research, and ranked in order of importance as numbers B-1 to B-10. Primary data were obtained through structured and semi-structured interviews and focus group discussions. The importance of numbers 1–10 in the ranking of the variables represents the weight of the score or comparison.

Second, through the above discussion on the trend of residential building continuity research, it is well known that evaluating the continuity of architectural features uses the spatial phrase, gamma, cluster, and variance methods. The spatial sentence method focuses on plane deconstruction. In contrast, the gamma analysis method can compare the similarity between the planes of traditional and modern houses after an investigation, using the spatial sentence method. In addition, combining the cluster analysis and variance method allows the similarity of other elements to be compared. The architectural continuity scores of residential buildings can be quantified using questionnaires and expert ratings, and scores were matched based on the importance of previously quantified behavioral variables. The questionnaire design was combined with a Likert scale, and the results were mainly composed of offline and online components to ensure comprehensive data acquisition. We analyzed the results for different age groups to select the final questionnaire, and aggregated the questionnaire scores with expert ratings. The degree of continuity between the architectural characteristics of traditional rural and contemporary residential buildings was derived by assessing the variables of both architectural features and user behavior, and quantifying specific numeric results in the preparation for the next stage of total score classification [17].

Finally, continuity scores in terms of architectural features and user behavior were summarized, and the degree of continuity of residential buildings was broken down into four ABCD grades by expert consultation, from the highest to the lowest. A level is defined as the continuity level of residential buildings with reasonable continuity, B level indicates that there are some problems in the continuity of residential buildings with room for improvement, and the residential buildings listed as C level suggest that there are serious problems. Rectification should be conducted according to the continuity variables of residential buildings. D-grade residential buildings indicate a loss of continuity. The constructed evaluation framework compares the continuity between traditional and new housing such that higher ranks imply better preservation of regional features that are important in shaping the value of architectural uniqueness and maintaining the diversity of architectural culture. In this study, a framework was constructed to assess the continuity of residential architecture and to understand the continuity of residential functions, types, and archetypes as dynamic and systematic (Figure 3). Based on this, a systematic evaluation framework combining architectural characteristics and user behavior was constructed to evaluate the design schemes of the residential buildings that have been put to use, and the new contemporary residential buildings or those that will be put into construction to ensure the continuity of traditional residential buildings in various regions (Figure 4).

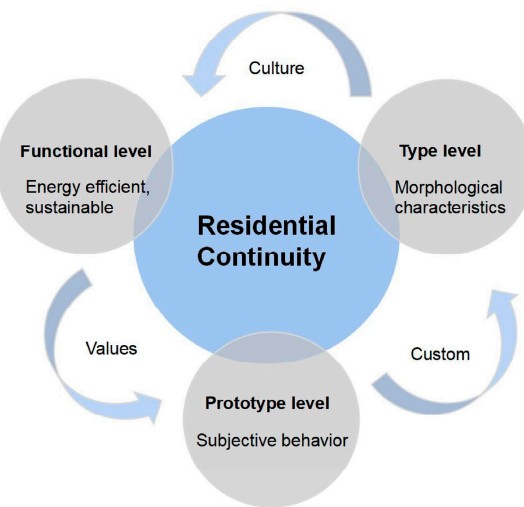

**Figure 3.** Continuity of residential architecture.

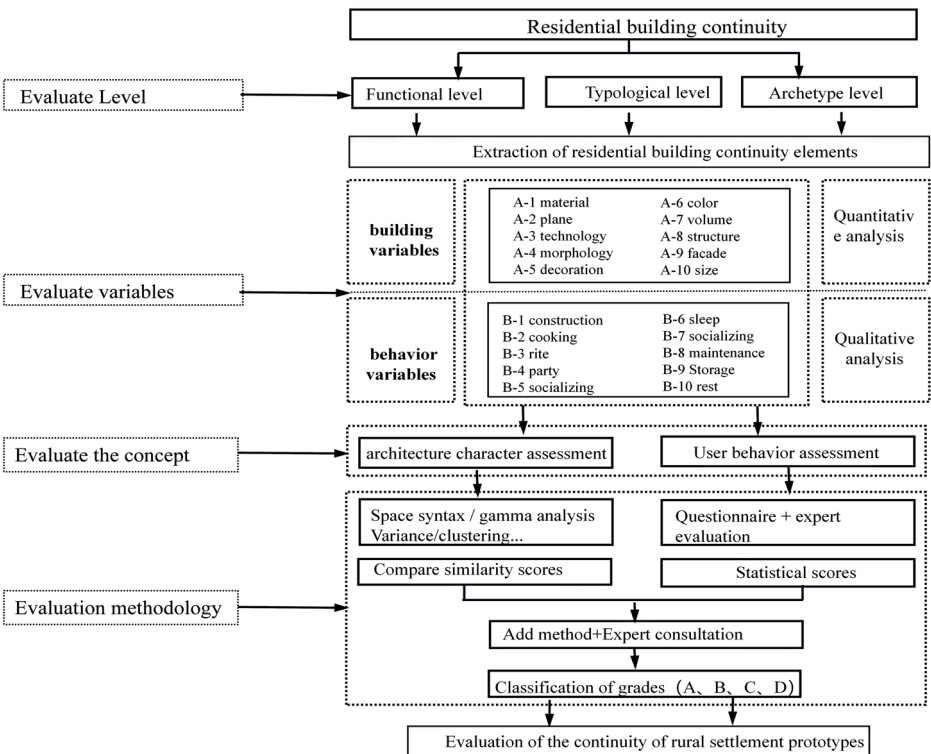

**Figure 4.** Continuity of the residential architecture assessment framework.

## 5. Conclusions

Today, traditional houses, which are an important part of the country's rural settlement heritage, are increasingly surrounded by typical urban dwellings, which are creating difficulties in the continuation of traditional houses and the transmission of culture. This study aimed to conduct a systematic literature review of the continuity of residential architecture in rural settlements and obtained 40 articles on the subject of residential architecture continuity by searching and filtering three databases (peer-reviewed and highly authoritative): WoS, Scopus, and EI. Through a review of those 40 papers, we found that the critical problem in the study of the continuity of residential architecture in rural settlements is the lack of focus on the behavior of residential users. Research on residential building continuity often begins with the objective aspect of residential building functions and the type and study of building materials, plans, technology, and other elements, although it

involves the archetypal level including the culture, values, customs, and other elements that affect the behavior of users. However, these studies focused on the objective elements of buildings and the environment. Therefore, an evaluation framework should be built using a human-centered approach that focuses on the study of user behaviors [73]. During the construction of the evaluation framework, we also quantified the research variables involved in the 40 residential building continuity articles, divided them into two levels, user behavior and architectural characteristics, and selected ten variables each from the levels of user behavior and architectural attributes according to the frequency of different variables. A range of research methods, such as the spatial sentence method, gamma analysis, cluster analysis, and the methodological method, which have been used often in previous studies, were combined to assess the characteristic building variables and user behavior variables. In addition to a framework for assessing the continuity of residential buildings in rural settlements based on function, levels of residential building types and prototypes, and the combination of architectural features and user behavior was constructed.

Inheritance of traditional homes requires an effort to establish a concept of "continuity" during the building stage [1]. Combining the features of architectural forms from traditional residential architecture and new materials reduces the damage caused by foreign architectural structures to the unique architectural style of rural settlements. This ensures harmony in rural landscapes. Furthermore, while inheriting the regional features of traditional residential architecture, more attention should be paid to the study of residents' behavior and routines. The development of an assessment framework for the continuity of residential buildings can help planners and designers to evaluate the continuity of residential buildings, develop regional planning principles, and implement design plans more efficiently.

This systematic review has some limitations. In terms of research content, the study of the continuity of rural settlement dwellings is mainly influenced by Western urban morphology. Previous studies have primarily focused on the objective architectural level, with less focus on the level of user behavior. Regarding the research methodology, we selected only three peer-reviewed, highly authoritative databases in the screening phase, and restricted the language to English, which may have resulted in missing some search results. We also included only high-quality reports with well-cited references [24], and excluded notes, webpages, and unpublished paper sources. Objective factors limited the systematic review conducted in this study.

Studying the continuity of residential buildings in rural settlements involves many theories in Western urban studies. As part of our future research, we plan to apply theories such as morphogenesis and typology to the study of rural residential architecture, which can potentially broaden the scope of theoretical inquiry to some degree. Furthermore, we plan to conduct an empirical investigation into the construction of a residential building continuity assessment framework to continue improving the hierarchical evaluation rules of the evaluation framework in the process of evaluating the continuity of residential buildings in different rural villages, in order to test the practicality of the evaluation framework for residential building continuity in rural settlements.

**Author Contributions:** Conceptualization, X.W. and L.Z.; methods, X.W.; software, X.W.; validation, X.W., J.L., N.Z., Y.T., Y.S., H.W. and C.C.; formal analysis, X.W.; investigation, X.W.; resources, H.W. and C.C.; data collection, Y.S. and Y.T.; writing—original draft preparation, X.W.; writing—review and editing, X.W., L.Z. and J.L.; visualization, X.W. and N.Z.; supervision, L.Z. All authors have read and agreed to the published version of the manuscript.

**Funding:** The Continuity of Residential Architecture Research Program was funded by the Major Program of the National Social Science Foundation (19ZDA191), Key Program of the Social Science Foundation of Hunan Province (21ZDB003), and High-end Thinking Tank Program of Central South University (2022znzk09).

**Data Availability Statement:** The data used to support the findings of this study can be made available by the corresponding author upon request.

**Conflicts of Interest:** The authors declare no conflict of interest.

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
