# Peer review of "Architectural Continuity Assessment of Rural Settlement Houses: A Systematic Literature Review"

_land, doi:10.3390/land12071399_

Round 1

Reviewer 1 Report

Thank you for giving me the opportunity to review this work. This is a very interesting article, well structured and written. I have only few comments/questions as follows:

-Since the aim of the article was ' to investigate the research trends, research categories, 15 and research variables through a systematic literature review. ' , could it be possible to actually present more extensively the research trends and discuss them in the discussion section? I feel that the article is mostly discussing definitions and other aspects but from a given perspective (e.g. Coates framework) without letting the data speaks. For example, many reviewed articles concerned heritage and development, two topics that could have been presented and discussed (e.g. is there a different perspective from western or non-western countries? Across the period of the studies, was there a change in developing the traditional residence to sustain its continuity? etc).

-In the same way, I am a bit surprised that there is a lack of discussion regarding the relationship between building types and morphology. Could it be expanded? For example, in the same way that this study showed there was a lack of focus on residents behaviour, I would like to know whether the reviewed literature embedded different scales while studying continuity and, if not, it could be a reason explaining the failure of policies to sustain continuity?

Introduction: could it be possible to explain the aim of this study (as it is explained in the abstract)?

Methodology:

-l107-112: I don't understand this section and I think it might be a problem arising from the translation. Table 1 shows keyword but this section talks about phrase. If there was a use of sentences, can you let us know which ones exactly?

-when was the start point of the literature review if 2022 was the end year? 2011? and why this choice?

l159-160: 'the continuity and inheritance of traditional residential architecture 'could it be possible to know if you used a time limit or not to consider what is a traditional residential architecture? For example, could you consider villages developed in the 1960s being traditional residential architecture?

Few English corrections to undertake:

-Reword the following sentences: line 106

-l121: not word but sign wildcard '*'

-l214: book title does not need capitalised letters

-l241: avoid 'a great deal of work' (too oral language)+ can you add evidence for this statement?

Excellent

Author Response

We thank the reviewers for their valuable comments and criticisms. Following the editor's and reviewers' suggestions, we have carefully revised the manuscript, and all revisions are highlighted in red in the revised manuscript. Below are detailed responses to all comments, criticisms and changes we have made. Please refer to the attachment. "Please let us know if you have any further questions. Thank you very much for your consideration.

Reviewer 2 Report

Dear Authors;

Thank you for letting me review your paper. It is a very interesting topic and I can see that it represents a lot of work on your park.  My comments below are meant to help you, even though they may seem very critical. I am guessing that English may not be your first language. This always makes it difficult when trying to compose a paper. I want to start with a couple general comments and then list some small changes in the text. I will also upload the PDF file of the paper that you wrote. I have highlighted different lines and sections in the paper. With each highlight there’s a little box that can be clicked on to see my comments. If you open this file in the Adobe Acrobat program, you should be able to see my highlights and comments.

My biggest concern is in the way that the paper is written. It needs an introduction that includes three things: (1) it tells the reader what the paper is about, (2) why the topic is important or why the reader should care about the topic and (3) how you will accomplish it or tells the reader what’s ahead.  This “what, why and how” are important throughout the paper. You usually do a good job of telling what you’re doing but very seldom do you provide an explanation (why) or rationale for what was done.  This makes the paper very difficult to understand. The reader is constantly struggling to know where you are at or where you were going, and why. I think this is particularly true in trying to understand why this is an important topic. What are the practical implications of the research?  Is there a practical problem and a research problem? I suggest that you look at The Craft of Research by Booth.  Your paper should tell a story that the reader can understand.

It looks like you've used some very useful tools and analyzing literature but you provide very little explanation of why these tools are good. For example, why did you choose the databases you used? Why did you narrow the number of articles down to 40.  How does that help improve your research? It may seem to the reader that in reducing the number of articles from 1140 to 40 that you might miss something.  The reader needs to know why you did that. It would give credibility to your work if you list the 40 titles.  Providing a citation helps give credibility to your research, but it does not necessarily help the reader understand why you did what you did.  In fact, I am concerned that some of your citations don’t seem to completely represent your research.  How did you get to the three key words used in the assessment of your literature data base.  Three seems very limited number – could you miss something? 

It is not clear what the objective of your paper is.  It seems that what you end up with is the process for assessing continuity, as depicted in Figure 7.  However, this is not clear to the reader until they get to the end of the paper. It would be helpful to the reader to have a clearer sense of what was being done.  Even the term continuity seems confusing. On the one hand, you acknowledge that real communities will change, but it seems that what you’re measuring doesn’t take that into account. At least it needs to be explained more completely.  Is more continuity better and why?  How much change is okay?

You use certain terms that while they may sound more sophisticated, don’t make the paper easier to understand I would suggest that you look at some of the terminology and see if you can make it easier to understand. In addition, at times, it seems as if you’re biased in your research, that you are looking at an outcome that favors continuity of rural settlements. You should try in your writing to appear to be more objective, and not to appear to be biased. It will give more credibility to your final result. It is not clear why you use the terms subjective and objective. They don’t seem to be as descriptive as other terms might be in the front and the use of these terms opens up a lot of questions about the phenomena that you’re looking at. While objective variables are easier to measure, that may not represent the reality of the phenomena. It would seem to me, that you should find some more accurate descriptive terms.

Tables 2,3,4,5 and 6 are unintelligible.  This table format is not a good way of showing graphically what you’ve done. I applaud your attempt to use some new graphic styles, but if they are not able to be understood, they’re not good.

There are typos in lines 99, 197, 200 and 216.

This is an important topic.  It would not do the topic justice without rewriting the paper so that it will be more understandable to the reader.  I wish you the best in this endeavor.

(See my to the authors)

Author Response

We thank the reviewers for their valuable comments and criticisms. Following the editor's and reviewers' suggestions, we have carefully revised the manuscript, and all revisions are marked in red in the revised manuscript. Below are detailed responses to all comments, criticisms and changes we have made. Please refer to the attachment. In addition, based on your suggestion, we have edited this article in English and attached an editorial report. Let us know if you have any other questions. Thank you very much for your consideration.

Reviewer 3 Report

Dear authors, 

Thank you for submitting the paper to Land. I enjoyed reading it and the methodological aspects for the literature review exercise you have done are quite interesting and well structured. However, I have a range observations (some more substantial than others) that I would suggest addressing to improve the sceientific stand of the paper. I am attaching a PDF file with this. 

I would consider the observations as important but possible to be addressed by you. In that sense, I would be OK to see a polished version of this manuscript before publication. I have clicked on 'minor revisions' as the option 'mid-revisions' does not exist as a chance. However, labelling the observations as 'minor' in this case does not mean the paper is 'almost ready' to be published. Please, don't misinterpret my veredict. 

Regards, 

Author Response

We thank the reviewers for their valuable comments and criticisms. Based on the suggestions of the editors and reviewers, we have carefully revised the manuscript, and all changes are highlighted in red in the revised manuscript. Below is a detailed response to all comments, criticisms, and changes we have made." Please see the attachment. "If you have any additional questions, please let us know. Your consideration is greatly appreciated.

Reviewer 4 Report

 The manuscript systematically organizes knowledge about architectural continuity assessment of rural settlement houses. The number of analyzed articles (883) is high. I believe that this manuscript will provide valuable information for researchers in this field. The manuscript is well organized, the methodology is clear, the results are presented in a clear way.

With regard to the manuscript, the following comments occur to me:

- in the Introduction section, in the last paragraph (lines 69-83) at quantitative analysis methods literature references are given, and at qualitative analysis method there are no such references - this should be completed,

- in section 2.1 Keyword Selection line 172 there is mention of experts - who were they?

- titles of tables and figures should not be separated from them (e.g., line 176 the title of Table 3 is on a different page from the table).

I believe that the manuscript deserves to be published in a version that takes into account the minor corrections indicated. I hope that these comments will contribute to raising the scientific level of the manuscript.

Author Response

(The authors gave the same response as above.)

Round 2

Reviewer 2 Report

Well done.  The rewritten versions tells not only what was done, but explains how and why it is important.  I know the authors put a lot of work in this, but I think it is worth it.  It makes a substantial contribution to the literature on the topic.  Again, well done.

check line 40 to see if wording is correct

check dates on line 150 and 153.  It reads like it continues before it began.